# Development of Composite Sponge Scaffolds Based on Carrageenan (CRG) and Cerium Oxide Nanoparticles (CeO_2_ NPs) for Hemostatic Applications

**DOI:** 10.3390/biomimetics8050409

**Published:** 2023-09-04

**Authors:** Kimia Alizadeh, Yasaman Dezvare, Shirin Kamyab, Jhaleh Amirian, Agnese Brangule, Dace Bandere

**Affiliations:** 1Department of Life Science, Faculty of New Sciences and Technologies, University of Tehran, North Kargar Street, Tehran 1439957131, Iran; alizadehn38@gmail.com (K.A.); yasidezi@gmail.com (Y.D.); shirin.kmb@gmail.com (S.K.); 2Department of Pharmaceutical Chemistry, Riga Stradins University, Dzirciema 16, LV-1007 Riga, Latvia; agnese.brangule@rsu.lv (A.B.); dace.bandere@rsu.lv (D.B.); 3Baltic Biomaterials Centre of Excellence, Headquarters at Riga Technical University, LV-1658 Riga, Latvia

**Keywords:** carrageenan (CRG), cerium oxide (CeO_2_), nanoparticles (NPs), composite, antibacterial properties, hemostatic effect

## Abstract

In this study, a novel absorbable hemostatic agent was developed using carrageenan (CRG) as a natural polymer and cerium oxide nanoparticles (CeO_2_ NPs). CRG-CeO_2_-0.5 and CRG-CeO_2_-1 composites were prepared by compositing CeO_2_ to CRG + CeO_2_ at a weight ratio of 0.5:100 and 1:100, respectively. The physicochemical and structural properties of these compounds were studied and compared with pristine CRG. Upon incorporation of CeO_2_ nanoparticles into the CRG matrix, significant reductions in hydrogel degradation were observed. In addition, it was noted that CRG-CeO_2_ exhibited better antibacterial and hemostatic properties than CRG hydrogel without CeO_2_ NPs. The biocompatibility of the materials was tested using the NIH 3T3 cell line, and all samples were found to be nontoxic. Particularly, CRG-CeO_2_-1 demonstrated superior hemostatic effects, biocompatibility, and a lower degradation rate since more CeO_2_ NPs were present in the CRG matrix. Therefore, CRG-CeO_2_-1 has the potential to be used as a hemostatic agent and wound dressing.

## 1. Introduction

Prehospital trauma deaths are largely caused by uncontrolled hemorrhage, whether it occurs in combat or in civilian settings. When such situations arise, materials that stop bleeding and dressings that enhance wound healing are required. Studies have shown that the application of hemostatic dressings can reduce mortality rates and speed up wound healing. Besides their capability of stopping bleeding rapidly, they are also capable of protecting against infection. Various materials have been reported to possess favorable characteristics in bleeding management. A variety of hemostatic materials are commercially available, including zeolite powder (ZLP) [1], silica [2], montmorillonite [2], kaolin [3], oxidized cellulose [4], collagen [5], gelatin [6,7], chitosan [8,9], silk fibroin [10], calcium alginate [8], fibrin [11], starch [12], peptides [13], polyethylene glycol [14], cyanoacrylate [15], and so on. Although these materials are hemostatic, they also have several limitations, including being expensive, having limited hemostatic or antibacterial effects, and being poorly biodegradable. Therefore, the development of safe and effective new hemostatic materials is essential to overcoming the disadvantages discussed above [16]. Recently, there has been increasing interest in hydrogels made from natural biopolymers, in particular polysaccharides, since they are biodegradable, compatible with living organisms, renewable, and safe [17]. They comprise a three-dimensional (3D) structure that holds a high water content, thus providing a suitable moist environment for wound healing [18,19]. Carrageenan (CRG) is a polysaccharide derived from red marine algae known as Rhodophyta [20]. It contains long linear chains of D-galactose and D-anhydrogalactose, along with anionic sulfate groups (–OSO_3_^−^) [20].

There are many types of antibacterial agents available, including antibiotics, nanoparticles (metal oxides and light-induced antibacterial agents), cationic organic agents, and others, which are applied to combat bacterial infections [21]. Antibiotics are the most commonly used method of treating this condition in clinical practice. However, antibiotic resistance is a concern that must be addressed [21,22,23]. Therefore, the growing prevalence of antibiotic-resistant bacteria strains in combat trauma wounds poses a growing challenge to clinicians [21]. Thus, the need for other antimicrobial agents to combat antibiotic resistance is apparent [24]. Several emerging fields, including nanomaterials, may be able to offer solutions to this problem since nanomaterials, in some cases, can be used to kill bacteria more effectively than conventional antibiotics [24].

Recently, nanomaterials have attracted considerable attention due to their unique properties resulting from higher surface-to-volume ratios [25]. Nanomaterials possess exceptional capabilities in terms of killing bacteria, compatibility with biological systems, and the ability to promote blood clotting [25]. Currently, cerium oxide nanoparticles (CeO_2_ NPs) are considered highly promising for wound healing applications due to their exceptional properties as antioxidants, anti-inflammatory agents, antibacterial agents, and stimulators of angiogenesis [26]. Their incorporation into different types of wound dressing, such as films, hydrogels, electrospun mats, sponges, and foams, could be considered an effective way of enhancing the antibacterial effect and controlling infection [27]. Moreover, their compatibility with biological systems and ability to promote clotting make them an ideal choice for treating combat trauma wounds. The combination of hydrogels and nanomaterials may be an effective dressing material for hemostatic and wound healing purposes. It is imperative to develop materials that are able to retain large amounts of water as well as have antimicrobial properties, blood clotting properties, and biological compatibility.

A variety of composite materials have been prepared using CRG in conjunction with other types of NPs or CeO_2_ NPs in conjunction with other hydrogels. Zhao et al. developed an alginate (ALG) hydrogel that contained different amounts of green-synthesized CeO_2_ NPs to enhance wound healing. NPs of CeO_2_ were synthesized with curcumin as a capping agent, and ALG-CeO_2_ composites were crosslinked with calcium chloride as the crosslinking agent. The researchers have provided evidence to support the efficacy of these composites in promoting wound healing, suggesting their potential as highly effective wound dressing materials utilizing hydrogels [28]. A study conducted by Yue-Hua Chen et al. in 2021 investigated the use of Gelatin methacryloyl (GelMa) hydrogel for diabetic skin and wound healing that contained various amounts of cerium-containing bioactive glass (Ce-BG) [29]. Composite GelMA/BG hydrogels containing 5 mol% CeO_2_ NPs exhibit strong antibacterial properties [29]. Further, it has been demonstrated that the composite hydrogel significantly improved wound healing by stimulating granulation tissue formation, collagen deposition, and angiogenesis in diabetic rats [29].

Therefore, our study aimed to synthesize a novel CRG-CeO_2_ composite biomaterial that was crosslinked using glutaraldehyde (GA). The CRG-CeO_2_ composite exhibited suitable physicochemical properties, biocompatibility, hemostatic effect, and antibacterial behavior, suggesting that it could be used as a promising hemostatic and wound-healing agent in the future.

## 2. Materials and Methods

### 2.1. Chemical and Reagents

Calcium chloride hydrate and κ-carrageenan were purchased from Sigma Aldrich (St. Louis, MO, USA). Cerium oxide nanopowder (CeO_2_ NPs) was bought from US Research Nanomaterials. Glutaraldehyde 25% solution in water was supplied from Titrachem. HCl fuming 37% also was purchased from Merck (Rahway, NJ, USA). The chemicals used were all analytical grade.

### 2.2. Methods

In order to prepare carrageenan (CRG) sponges and carrageenan-cerium oxide nanoparticles (CRG-CeO_2_) sponges, 0.025 g of CRG powder was added to 5 mL of deionized water. Afterwards, the solution was placed into an oil bath at 70 °C for 30 min until completely dissolved. In subsequent steps, various amounts of CeO_2_ NPs—0, 0.5, and 1 wt.% (relative to CRG)—were dispersed in deionized water using a sonicator and separately added to the CRG solutions and stirred in the dark, as shown in Table 1. The pH of the solutions was subsequently adjusted to 3, and 100 µL glutaraldehyde (GA, 10% volume ratio) was added to the solution and stirred overnight. In the end, the solutions were poured onto a plate, placed in a −20 °C freezer for 24 h, and then frozen for 48 h, as shown in Figure 1.

### 2.3. Characterization

#### 2.3.1. Scanning Electron Microscopy (SEM)

The surface morphology images of CRG and CRG-CeO_2_ composite scaffolds were captured using a scanning electron microscope (FESEM, MIRA3 FEG-SEM, Tescan, Brno, Czechia) operating at 20 kV. Prior to characterizing the samples, they were all sputter-coated with gold. In order to conduct SEM analysis of CeO_2_ NPs, a small quantity of NPs was disseminated in ethanol and subjected to sonication in an ice bath. Subsequently, the resulting suspension was deposited onto an aluminum substrate and allowed to air dry. ImageJ 1.52v software was utilized to determine the pore size of each group. Photos were taken from ten randomly selected areas of each sample.

#### 2.3.2. X-ray Diffraction Analysis (XRD)

X-ray diffraction analysis (XRD) was performed using an X-ray diffractometer (Inel, Artenay, France).

#### 2.3.3. Fourier Transform Infrared (FTIR) Spectroscopy

FTIR analysis was carried out using a Fourier transform infrared spectrophotometer (Bruker vector 33, Mannheim, Germany) in the range 400–4000 cm^−1^.

#### 2.3.4. Swelling and Degradation

The swelling and degradation behaviours of CRG and CRG-CeO_2_ composite scaffolds were investigated by immersing them in PBS at 37 °C for designated time intervals (Appendix A). The effect of the incorporation of CeO_2_ NPs into a CRG matrix on the swelling and degradation behaviours was determined by gravimetric analysis. Initially, the dry weight of CRG and CRG-CeO_2_ hydrogels was recorded as *W*0. Next, 1 mL of PBS was added to the sample and allowed to incubate at 37 °C. Within designated time intervals (1 h, 2 h, 9 h, and 24 h), the weight of swelling hydrogels was measured until equilibrium was reached. Based on Equation (1), the swelling percentage was determined.
(1)$$Swelling\;ratio\;\%=\frac{Wt-W0}{W0}\times 100$$
where *Wt* represents the weight of the swollen sample at a predetermined time, while *W*0 represents the weight of the sample at the beginning of the study.

The degradation of the CRG and CRG-CeO_2_ composite scaffolds was performed according to ASTM F1635-16, which is widely used in order to assess the degradation of hydrogels. To determine how the hydrogel degraded in PBS at 37 °C, the weight change of the sample was determined in intervals of 1, 3, and 5 days. A degradation percentage was calculated using Equation (2).
(2)$$Degradation\;\%=\frac{W0-Wt}{W0}\times 100$$
where *Wt* represents the weight of the degraded sample at a specific time, while *W*0 represents the weight of the sample at the start. Both swelling and degeneration were assessed using three replicates.

### 2.4. Whole-Blood Clotting Index

The determination of the blood clotting index (BCI) was performed according to a method that had been used previously [30]. Recalcified whole blood solution was prepared by adding 8 µL of calcium chloride (CaCl_2_) to 100 µL of blood at a concentration of 0.2 M. The CRG, CRG-CeO_2_-0.5, and CRG-CeO_2_-1 scaffolds, as well as the gauze (Zarrin Teb, Kerman, Iran) as a positive control, were cut into cylindrical sections (5 mm × 8 mm) and 50 µL of recalcified whole blood solution was added to them. The samples and the positive control were incubated at various intervals of 30 s, 60 s, 90 s, and 120 s. Subsequently, 10 mL of deionized water was carefully added so as not to disturb the clot, and 8 mL of the solution was centrifuged at 3000 rpm for one minute. In the next step, the supernatant was placed in an incubator for one hour. After this time, the absorbance of 2 mL of the samples and the control was determined using a UV-Vis spectrophotometer at a wavelength of 540 nm.

### 2.5. Antibacterial Study Using the Disc method

Agar disk diffusion was used to determine the antibacterial activity of the CRG, CRG-CeO_2_-0.5, and CRG-CeO_2_-1 scaffolds. A study was conducted on two bacteria, including Gram-negative *Escherichia coli* (*E. coli*, ATCC 9637) and Gram-positive *Staphylococcus aureus* (*S. aureus*, ATCC 12600). As a first step, 14 g of nutrient agar (QUELAB) was suspended in 500 mL deionized water and stirred for 15 min to obtain a medium culture. Following this, the agar solution was autoclaved for two hours at 121 °C. Upon completion of this procedure, the solution was poured into a 10 cm Petri dish. Following solidification, fresh cultures of the bacteria *E. coli* and *S. aureus* were smeared evenly on the agar medium using sterile swabs. CRG, CRG-CeO_2_-0.5, and CRG-CeO_2_-1 samples were placed aseptically on the agar surface and incubated for 24 h at 37 °C. The inhibition zones for each sample, if present, were determined and documented after the incubation time. As a positive control, vancomycin antibiotic discs (30 mg, Padtan Teb Company, Qods, Iran) were used.

### 2.6. Cytotoxicity Study

NIH 3T3 cells were obtained from the National Cell Bank of Iran (NCBI, Tehran, Iran). An RPMI-1640 medium (Gibco, Darmstadt, Germany) supplemented with 10% Fetal Bovine Serum (FBS) and 100 IU per milliliter streptomycin (Gibco, Darmstadt, Germany) was used for cell cultivation.

An MTT assay was conducted to determine the viability of cells on the CRG, CRG-CeO_2_-0.5, and CRG-CeO_2_-1 scaffolds. A minimum of three replications were conducted for each type of sample (each sample weighs approximately 0.01 g). Following previous studies [31], the samples were sterilized by soaking in 70% ethanol for 60 min and then washing three times in PBS. A hydrogel extract was obtained by immersing the sterilized samples in 200 µL of extract of the scaffold in RPMI-1640 culture medium. NIH 3T3 fibroblast cells were seeded in a 96-well plate at a density of 1 × 10^4^ cells per well. Seed cells were treated for one and three days with an extracted medium derived from sterilized hydrogel—RPMI-1640 medium containing 10% FBS. After 1 and 3 days, 10 μL of MTT (Bio Idea Co., Noavari, Iran, with a concentration of 5 mg/mL) was added to the wells, followed by an incubation period of 4 h at 37 °C. Afterwards, the solutions were removed from the wells and 100 μL DMSO was added to each well, allowing the crystals to dissolve for one hour. The absorbance of the solution was determined at 570 nm with a plate reader.

### 2.7. Statistical Analysis

Graph Pad Prism 8 (Graph Pad Software Inc., San Diego, CA, USA) was used to perform all statistical analyses. Analyses were conducted using one-way or two-way ANOVA, depending on the experimental design. The results were expressed as mean ± standard deviation. The significance levels were defined as * *p* < 0.05, ** *p* < 0.01, and *** *p* < 0.001.

## 3. Results

### 3.1. Nano- and Micro-Structure and Material Properties of the Sponges

Figure 1 shows surface and cross-section SEM images of freeze-dried CRG, CRG-CeO_2_-0.5, CRG-CeO_2_-1 scaffolds, and CeO_2_ NPs. Figure 1D and Appendix A shows that the CeO_2_ NPs are in the nanoscale and have a size distribution from 15–30 nm. According to low-magnification images, all samples appear to be connected and well structured. Additionally, high-magnification images of both CRG-CeO_2_ composite scaffolds indicate that CeO_2_ NPs were well distributed within CRG hydrogels. Depending on the type of sample, the pore size range varied between 40 and 200 μm. This figure illustrates that pristine CRG hydrogel has pores ranging from 60 to 190 μm, with an average size of 135.69 ± 29.3 µm. However, as can be seen, the addition of 0.5% and 1% CeO_2_ NPs significantly reduced the average pore diameter of the CRG-CeO_2_-0.5 and CRG-CeO_2_-1 composite scaffolds, respectively, to 103.02 ± 30.04 µm and 60.5 ± 12.8 µm compared to the pristine CRG. This is attributed to the electrostatic interaction between negatively charged CRG polysaccharides and positively charged CeO_2_ NPs, which is consistent with previous findings [32,33]. By increasing CeO_2_ NPs from 0.5 to 1%, the hydrogel became more compact, and pores became smaller. Furthermore, CeO_2_ NPs may function as crosslinking agents, so altering the amount of CeO_2_ NPs can alter the degree of crosslinking. Eventually, this interferes with the growth of ice crystals during the freezing stage, resulting in a reduced pore size on the scaffolds when they are frozen.

As shown in Figure 1E–G, energy-dispersive X-ray spectroscopy (EDS) analysis reveals the presence of C, O, S, and C, O, S, Ce elements in CRG and both CRG-CeO_2_ composite hydrogels, respectively. Figure 2 shows EDS mapping for the CRG-CeO_2_-0.5 and CRG-CeO_2_-1 scaffolds. According to the EDS mapping that was performed for both the CRG-CeO_2_-0.5 and CRG-CeO_2_-1 composite hydrogels, CeO_2_ NPs were distributed evenly throughout them.

As shown in Figure 3A,B, the crosslinking of CRG and CRG-CeO_2_ hydrogels occurred through a reaction between the hydroxyl group of CRG and the aldehyde group of GA. Predicted mechanisms of the crosslinking reaction are in good agreement with previous studies [34]. An FTIR spectroscopy analysis was performed to examine the chemical interaction between CRG and CeO_2_ NPs as well as their chemical functionality. Figure 3C shows the FTIR spectrum of CRG, CRG-CeO_2_-0.5, and CRG-CeO_2_-1 scaffolds. It can be seen that scaffolds containing CRG exhibit a board band in the 3200–3400 cm^−1^ region due to stretching vibrations within O-H groups and hydrogen bonds [35,36]. Based on Figure 3C, it is apparent that the peak at 3200 cm^−1^ to 3600 cm^−1^ of width becomes a boundary and intensifies, which is attributed to the formation of both intramolecular and intermolecular hydrogen bonds [36]. Furthermore, two peaks at 2959 cm^−1^ and 2850 cm^−1^ were observed, which are related to the C-H stretching bands [37,38]. Moreover, ester sulfate (O=S=O) symmetric vibrations occur at 1220 cm^−1^, and -O-SO_3_ stretching vibrations occur at D-galactose-4-sulfate (G4S) at 844 cm^−1^ and D-galactose-2-sulfate (DA2S) at 801 cm^−1^, respectively [39,40,41,42]. All samples containing carrageenan, including CRG, CRG-CeO_2_-0.5, and CRG-CeO_2_-1 exhibited these peaks [39,40,41,42]. Moreover, changes in the 1090 cm^−1^ and 934 cm^−1^ regions would result in vibrations in the CC, COC, and OH groups in the CRG structure. There will also be demonstrations of the formation of hydrogen bonds between CeO_2_ NPs and CRG, as well as GA crosslinking [43]. In accordance with previous studies, the band at 848 cm^−1^ is associated with a metal–oxygen bond. It is apparent from the results that the addition of CeO_2_ NPs intensified peaks at 1066 cm^−1^ and 860 cm^−1^ due to the stretching vibration of Ce-O as well, which is specific to CeO_2_ NPs [36].

XRD was used to determine the crystallinity and phase composition of all CRG, CRG-CeO2-0.5, and CRG-CeO2-1 scaffolds. Figure 3D illustrates a broad peak in the CRG scaffold, a sample without CeO2 NPs, indicating amorphous structure of CRG. The introduction of CeO2 NPs into the CRG matrix resulted in the appearance of crystalline peaks at 2θ of 28.4°, 33.1°, 47.4°, and 56.2° specified to (111), (200), (220), and (311) crystallographic planes, respectively [44,45].

### 3.2. Swelling and Degradation

Figure 4A illustrates the dynamic swelling behavior of the CRG and CRG-CeO_2_ scaffolds in PBS. The weight of all samples increased within the first few hours of immersion. The weight then started to decrease and then started to level off after a certain period of time. It can be seen that incorporating CeO_2_ into CRG results in a decrease in swelling behavior when compared to neat CRG. This decrease in swelling behavior is attributed to the increased crosslinking density of the polymer matrix due to the incorporation of CeO_2_ NPs. This improved crosslinking leads to a more rigid polymer network with fewer voids, resulting in a decrease in swelling behavior. The SEM images in Figure 1 illustrate that CRG-CeO_2_-1 has a more compact structure with a smaller pore size as compared to CRG-CeO_2_-0.5 and CRG hydrogels. Thus, their initial amounts of PBS were reduced in total, which was in accordance with the swelling capability study shown in Figure 4A. The decrease in swelling behavior can potentially be attributed to the inherent fragility of hydrogels. Earlier studies reported that swelling behavior differed depending on the shape of the porous structure, as well as the type of crosslinking applied using inorganic compounds. There is good agreement between the results of our study and those reported in the literature.

Compared to CRG, both CRG-CeO_2_-0.5 and CRG-CeO_2_-1 exhibit a lower degradation rate and weight loss value, as shown in Figure 4B. This can also be attributed to the fact that the presence of more crosslinks in the hydrogel decreases the molecular weights between the crosslinks. Thus, this results in a reduction in the free volume between macromolecular chains, which, in turn, allows water molecules to penetrate the chain [46]. Figure 4B shows that after five days of incubation, the CRG-CeO_2_-1 hydrogels lost 20% of their original weight, which is about 50% less than the weight loss associated with the neat CRG scaffold after five days. The incorporation of CeO_2_ NPs into CRG matrix, with an increasing amount from 0.5 to 1, is attributed to the increased crosslinking density as well as to a decrease in weight loss during incubation.

Hydrogel-based wound dressings are a promising candidate for wound dressing, as they provide moist conditions that speed up the healing process. To facilitate the production of new ECM on the skin, scaffolds should be capable of absorbing water between 100 and 800 times their dry weight. Such a scaffold should also be able to maintain its shape and structural integrity in the presence of moisture, providing an effective substrate for cell adhesion and growth. The polysaccharides such as CRG [47], chitosan [48], pectin [49], etc. bind strongly to water, resulting in easy diffusion of encapsulated therapeutics from them. This results in a low loading capacity and a rapid release rate [50]. To address this issue, inorganic compounds are used to crosslink polysaccharides to reduce their permeability.

### 3.3. In Vitro Blood Clotting Evaluation

The blood clotting capability of materials was assessed using a dynamic whole-blood clotting test, where a solution with a higher absorbance value indicates a slower clotting rate. As a control group, gauze was utilized as a traditional hemostatic agent. Figure 5 shows that in comparison to gauze, all samples, including CRG, CRG-CeO_2_-0.5, and CRG-CeO_2_-1, displayed a lower BCI. CRG-CeO_2_-0.5 and CRG-CeO_2_-1, however, demonstrated lower BCI values than CRG, indicating that they have greater hemostatic properties than CRG. Accordingly, the introduction of CeO_2_ NPs into the CRG matrix resulted in a lower BCI rate and an improved clotting index rate in comparison to neat CRG. All samples and controls showed a downward trend in BCI as the incubation time was extended from 30 s to 120 s. The combination of all factors, including their highly porous structure, swelling properties, rapid blood adsorption, and excellent in vitro blood clotting effect, makes them ideal hemostatic materials.

Currently, CRG is used widely as wound dressings due to its ability to absorb pseudo-extracellular fluid [51]. Previous studies have shown that negatively charged surfaces in CRG can activate factor XII, initiating the intrinsic coagulation pathway [51]. Studies on mice showed that unmodified CRG had hemostatic properties comparable to Gelfoam (commercial hemostatic agent) [51]. The poor mechanical strength of this product makes it difficult to disintegrate after swelling and unable to carry body fluids [51]. As shown in Appendix A, crosslinking CRG with GA and incorporating CeO_2_ NPs into CRG resulted in improved stability, making CRG particularly suitable for hemostatic applications. The porous and three-dimensional architecture of the scaffold, as well as the coagulation capability of both CRG and CeO_2_ NPs, allow it to absorb and retain blood as well as prevent bleeding. Meanwhile, CRG contains high levels of hydroxyl groups, thus making them hydrophilic biomaterials that may be helpful in surface wetting and adhesion and consequently control bleeding [52,53].

### 3.4. Antibacterial Activity

The antibacterial properties of the composite scaffolds were investigated against *S. aureus* and *E. coli*. These two kinds of bacteria were chosen because they are among the top five pathogens associated with diabetic and non-diabetic wound infections [54]. Figure 6 shows the antibacterial response of CRG hydrogel with and without CeO_2_ NPs after 24 h of incubation with *E. coli* and *S. aureus* bacteria. Observations indicate that both CRG-CeO_2_-0.5 and CRG-CeO_2_-1 demonstrate antibacterial properties, whereas CRG does not show antibacterial properties. According to our findings, CRG-CeO_2_ hydrogels containing a higher concentration of CeO_2_ NPs demonstrated greater activity than hydrogels containing a lower concentration of CeO_2_ NPs.

The mechanism can be explained by the fact that positively charged NPs were adsorbed onto the membranes of negatively charged Gram-positive and Gram-negative bacteria as a result of electrostatic interactions. In this way, CeO_2_ NPs remain on the surface of bacteria for a long time rather than penetrating their membranes, resulting in the blockage of the membranes of bacteria. Following this, NPs may have a detrimental effect on the viscosity of the membrane, impair the specific ionic pumps, and cause a significant disruption in transport exchange between the bacterial cell and the solution [55]. As a second consideration, CeO_2_ NPs may be capable of attacking proteins after they bind to the outer membrane of a bacterial cell [55,56]. Moreover, the released cerium ions could disrupt electron flow and respiration by bacteria, react with thiol groups (-SH), or be absorbed by transporters and/or porins, thereby preventing the transport of nutrients to the bacteria. Studies demonstrate that CeO_2_ NPs can physically damage bacterial membranes, especially in Gram-positive bacteria, due to their irregular structure [55,57].

### 3.5. Cell Study and Cytotoxicity

An evaluation of the cytotoxicity of scaffolds made from CRG, CRG-CeO_2_-0.5, and CRG-CeO_2_-1 was conducted using the NIH 3T3 cell line. NIH 3T3 cells seeded in 96-well plates were treated with an extracted solution separately derived from the scaffolds. As shown in Figure 7, all samples exhibited viability greater than 85%, which indicates that the materials are biocompatible. Compared with the CRG scaffold, scaffolds containing CeO_2_ NPs, CRG-CeO_2_-0.5 and CRG-CeO_2_-1 demonstrated a higher degree of cell viability, which is in line with previous studies [44,58]. CeO_2_ NPs were released into the culture medium after 24 h and 72 h, leading to an increase in metabolic rate and decreased production of reactive oxygen species (ROS) [44,59]. However, CRG-CeO_2_-1 scaffolds showed an insignificant loss of cell viability on day 3, possibly due to the overexpression of pro-inflammatory factors that may affect cellular function [45]. Several studies have investigated CRG’s biocompatibility by using it as a hydrogel for encapsulation of human adipose stem cells (hADSCs) [60,61]. Furthermore, CeO_2_ NPs are known to be highly biocompatible, making them ideal for biomedical applications as well as wound healing [44,62]. An investigation was conducted by Kalantari et al. to determine whether CeO_2_ NPs in concentrations ranging from 0 to 1% are cytotoxic [44]. Based on their findings, there is no evidence that cell membranes are injured by CeO_2_ NPs at these concentrations, suggesting that CeO_2_ NPs at these levels have no toxic effects on cells [44]. CRG can also be utilized as a capping agent for the green synthesis of CeO_2_ NPs. Nourmohammadi et al. utilized CRG as a capping agent to prepare CeO_2_ NPs in an environmentally friendly manner [63]. NPs of CeO_2_ were produced in spherical shapes with an average diameter of 34 nm [63]. Synthesized CeO_2_ NPs with concentrations ranging from 0 to 500 µg/mL were evaluated for their cytotoxicity on WEHI 164 cells. In general, CeO_2_ NPs appeared to have a very low cytotoxic effect on WEHI 164 cells, which makes them suitable for a variety of biomedical applications [63].

## 4. Discussion

Multifunctional hydrogels could be suitable as wound healing agents [64]. CRG is one of the hydrogels that is extensively investigated for its potential as a hydrogel in biomedical research. This is a result of its exceptional hydrophilicity, superior biocompatibility, low cytotoxicity, biodegradability, and capacity to sustain nanomaterial release [65,66]. In terms of wound healing applications, CeO_2_ NPs show great promise, as they are capable of effective wound closure, tissue regeneration, and scar reduction [26]. Incorporating NPs into hydrogels holds significant potential for use as versatile wound dressings, offering a wide range of application opportunities [67]. CeO_2_ NPs exhibit great potential as a nanomaterial for wound healing purposes, as they possess the ability to promote wound closure, facilitate tissue regeneration, and minimize scarring. NP-incorporated hydrogels hold significant potential as versatile dressings for wound healing, offering wide-ranging application possibilities [6]. Various materials can be used as crosslinking agents for CRG hydrogels [68]. GA is commonly used as a crosslinking agent for crosslinking polymeric hydrogels [68]. Due to the complexity of wound healing, a wound dressing should address multiple challenges encountered during skin wound healing [69]. A key aspect when developing hydrogels for wound healing applications is to consider important factors such as antibacterial properties, moisture retention at the wound site, hemostatic properties, and enhancement of cellular behavior [69].

This study aimed to develop and create multifunctional biomaterials based on CRG and CeO_2_ NPs that have hemostatic properties, antibacterial properties, and high biocompatibility. To accomplish this, various weight percentages of CeO_2_ NPs—0, 0.5, and 1% (relative to CRG)—were incorporated into the CRG hydrogel. Based on SEM results, electrostatic interactions between positively charged CeO_2_ NPs and negatively charged CRG resulted in the formation of a compact hydrogel structure, with smaller pores, when a higher amount of NPs was added. Pores present in the biopolymer matrix contribute significantly to the healing process by facilitating cell filtration, promoting high permeability, and allowing oxygen and nutrients to diffuse [70]. Studies have shown that the optimal pore size in wound dressings in adult skin healing is between 20 to125 µm, providing a suitable environment for effective wound management [71]. Water retention is one of the major concerns associated with the use of hydrogel as wound dressings. If exudate is not absorbed adequately, infection may occur, and the dressing may separate from the wound. Contrary to this, if there is little exudation, the dressing will be more likely to dry out and be ineffective. Thus, it is imperative to consider swelling characteristics and water retention properties when evaluating dressing materials [72]. By incorporating CeO_2_ NPs into CRG, the swelling percentage was reduced, which may be attributed to a reduction in available void spaces between polymeric chains. Furthermore, the degradation rate is also decreased as a result of this reduction in free volume between macromolecular chains [48].

Local hemostatic agents have several benefits in the medical field. These type of agents effectively enhance blood preservation by minimizing fluid loss, expediting the process of hemostasis, limiting the potential negative consequences associated with systemic hemostatic medications, and saving valuable transfusion blood supplies. In recent years, significant efforts have been put into creating high-tech dressings that actively induce or promote hemostasis beyond just absorbency [73]. Chen et al. designed dendritic hydrogels to stop bleeding, kill bacteria efficiently, convert macrophages from proinflammatory to repaired, and increase collagen deposition and blood vessel formation [64]. Wound dressings with antibacterial characteristics are essential for preventing infections and fostering healing during wound care [74]. Metallic nanoparticles eliminate bacteria by adhering to their cell membranes, resulting in membrane disintegration, leakage of bacterial contents, and inhibition of protein synthesis [75]. Based on the BCI results, all samples demonstrated significantly superior blood clotting capability compared to commercial gauze. According to our findings, the CRG-CeO_2_ hydrogels demonstrate favorable characteristics for multifunctional applications and can serve effectively as hemostatic and wound healing agents while preserving structural integrity. In future investigations, a series of in vivo experiments will be conducted to further understand the effectiveness of hydrogels.

## 5. Conclusions

In this study, we produced CRG and CRG-CeO_2_ scaffolds by freeze-drying and crosslinking them with GA. SEM micrographs showed interconnected pores with pore sizes of around 136, 103, and 60 µm for CRG, CRG-CeO_2_-0.5, and CRG-CeO_2_-1, respectively, indicating a more compact structure after incorporating CeO_2_ NPs into the CRG matrix. Incorporating CeO_2_ NPs into the CRG matrix enhanced hemostatic performance, as demonstrated by the BCI. The BCI test indicated that CRG-CeO_2_-0.5 and CRG-CeO_2_-1 scaffolds have hemostatic activity comparable to CRG and gauze (the conventional hemostat in use in hospitals). There is evidence that CeO_2_ NPs enhance antibacterial activity in CRG-CeO_2_ hydrogels; however, CeO_2_ NPs concentration influences pore size, degradation, and swelling behaviors. The in vitro results also revealed that both CRG-CeO_2_-0.5 and CRG-CeO_2_-1 showed high cell viability and nontoxic behavior compared to the results obtained with CRG and control samples. According to these findings, the incorporation of CeO_2_ NPs into CRG hydrogels could lead to the development of hemostatic agents that are antibacterial in nature.

## Data Availability

Not applicable.

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
