# Peer review of "Development of Composite Sponge Scaffolds Based on Carrageenan (CRG) and Cerium Oxide Nanoparticles (CeO2 NPs) for Hemostatic Applications"

_biomimetics, 2023, doi:10.3390/biomimetics8050409_

Round 1

Reviewer 1 Report

In this study, the authors reported a novel absorbable hemostatic agent with good anti-bacterial activity prepared using carrageenan and cerium oxide nanoparticles by freeze-drying and crosslinking method. The authors investigated the microstructure and material properties, swelling and degradation behaviors, in vitro blood clotting index, antibacterial activity, and cytotoxicity of prepared CRG-CeO2  sponge. The presented results suggested CRG-CeO2 exhibited good physicochemical properties, biocompatibility, hemostatic effect and antibacterial behavior, and could be used as a potential hemostatic and wound healing agent.

The research work is interesting and promising.

Here are several questions and/or suggestions the authors may consider. 

1.    The section 4 is missing in the manuscript.

2.    Figure 3A and B are not mentioned in the text.

3.    There are some typos in the manuscript, such as "ESD" in line 204 should be "EDS", "Figure 3D" in line 217 should be "Figure 3C"; and "CeO2" in line 233-234 should be "CeO2", etc.

4.    The authors mention that this scaffold is able to maintain its shape and structural integrity in the presence of moisture. The authors should provide a figure to show the changes in the shapes of this prepared sponge before and after soaking with water?

5.    Water vapor transmittance and the passage rate of suitable gases such as O2 and CO2 are also meaningful indicators for evaluating hydrogels properties considering its potential application as hemostatic agent and wound dressing.

6.    My biggest concern is the evolution of cytotoxicity. Indeed, here the authors only showed that the prepared sponge is "stable" enough in culture medium and no toxic material will be dissolved into the medium to kill the mammalian cells. However, considering the proposed application (hemostatic agent and wound dressing), this sponge should directly contact the would area. Therefore, is it possible the sponge kill the mammalian cells directly? At least the authors should discuss the safety of CeO2?

Author Response

We would like to thank you the Reviewers and Editors for their detailed considerations of the paper, valuable inputs and constructive comments overall. We have addressed each point with our response outlined below in blue. All changes/additions to text and figures have been summarized below in red. Textual changes have been added to the manuscripts using tracked changes and in red.  

Reviewer 2 Report

At Methods there is a very large inconsistency between the quantities of substances used. Certain quantities are stated in the text, others in the table. How much was actually used?

Verify and correct CeO2 in all manuscript

How was the pore size determined/calculated?

Line 194 - what does "1,2" mean???

At the explanation of FTIR, there are 3 options: cm, cm-1, cm-1. Can you choose only one?

Line 221 - is caps lock necessary? - Stretching Bands

In the text, there is no reference to Figure 4B and Figure 5

Line 296 - a parenthesis must be closed

Author Response

(The authors gave the same response as above.)

Reviewer 3 Report

1. Page 1, line 16 CRG-CeO2-0.5 and CRG-CeO2-1 composites were prepared by compositing CRG and CeO2 at a weight ratio of 200:1 and 100:1,respectively”, Does not coincide with the mass ratio in the Table 1,And, on page 3 of the article, Line 90,“0.025 g of carrageenan powder was added to 5 mL of deionized water”, Here, the mass fraction is 0.5 wt.%, inconsistent.

2. Modify Figure 3 A. B) Schematic of the CRG and CRG-CeO2 scaffolds and their cross linking with glucose, (C) FT-IR, and (D) XRD spectrum of a) CRG, b) CRG-CeO2-0.5, and c) CRG-CeO2-0.5;

On page 8 and line 217 of the article, "Based on the Figure 3D, it is apparent that the peak at 3200 to 3600 cm of width benefits a boundary and intensities..." should be changed to "Figure 3C".

3. Antibacterial mechanism can be strengthened by citing 10.1016/j.cej.2023.141852; 10.1016/j.mtadv.2022.100271 and what are the advantages of the current work compared to published articles?

4. In Figures 3A and B, what does GA represent? There is no explanation in the entire text. If it is the glutaraldehyde mentioned in the introduction, is the structure accurate?

5. On page 9 of the article, in Figure 4A, why does the dynamic expansion behavior of the three samples show a downward trend after increasing in weight in the first few hours before stabilizing, rather than directly stabilizing?

6. On page 11 of the article, in line 336, "As shown in Figure 6, all samples exhibit viability greater than 85%...", the "Figure 6" should be changed to "Figure 7".

Author Response

(The authors gave the same response as above.)

Reviewer 4 Report

Although authors have presented a well defined and structured manuscript, the below mention suggestion can polish the draft and bring more visibility to readers  

1. Suggested to indicate the quantity of composite used to analyse swelling behaviours.

2. Supplement the details of gauze used for whole-blood clotting analysis.

3. What was the hypothesis of selecting only Escherichia coli and Staphylococcus aureus to mimic the pathogen condition of wound, their are other several potential pathogen that represent the class

4. It would more interesting if authors presents the cross sections SEM image of the Composite

5. Suggested to supplement the whole blood clotting image (possibly refer the paper: https://doi.org/10.1007/s10924-023-02819-9)

6. The zone of inhibition for E. coli against CRG-CeO2-1 needs to either reverified by performing the experiment or replace the figure as the zone almost crossed the section to other, which does not reflect the accuracy of the results.

7. What was the quantify of samples were used for MTT assay is not clear from method, suggested to indicated what was the weight before sterilisation using organic solvent and after. Interestingly why not long UV exposure was used? suggest to explain that hypothesis too.

Good Luck

Thanks

Author Response

(The authors gave the same response as above.)

Reviewer 5 Report

The paper reports on the fabrication of nanoceria-carrageenan porous scaffolds that are promising for wound healing applications. The subject of the paper fits the scope of Biomimetics journal.

I have the following comments:

In the Introduction section, I would suggest a more comprehensive discussion of the current examples of wound healing applications of nano-ceria. Please also provide the information on the use of ceria-polymer composites for wound healing.

The only existing paper concerning ceria nano particles incorporated into carrageenan is ignored (https://www.sciencedirect.com/science/article/abs/pii/S0272884218319436). Please provide a comprehensive discussion of the results presented in this paper and in the current paper.

In Section 2.3.2, please replace “spectrometer” with “diffractometer”.

Line 204: please replace ESD with EDS.

Please correct typos (Lines 84, 144, 216 etc.).

Please provide TEM images of CeO2 nanoparticles used in the study to confirm nano-size of the particles.

The action mechanism of ceria-carrageenan composites should be hypothesized taking into account already existing reports on the bioactivity of nano-ceria and carrageenan itself.

Moderate editing of English language is required

Author Response

(The authors gave the same response as above.)

Round 2

Reviewer 1 Report

The authors have addressed the comments very well.

Reviewer 4 Report

The authors have reflected the said suggestion and I recommend editor for acceptance in current.